Description of a novel Ligia species from Nihoa, a remote island in the Papahānaumokuākea Marine National Monument

Santamaria Carlos A. santamaria.carlos.a@gmail.com 1
Bork Annabelle 1
Larson Alexandra J. 1
Link Daniel J. 2
1 Department of Biology, University of Tampa , Tampa , FL , United States of America
2 Papahānaumokuākea Marine National Monument, US Fish and Wildlife Service , Honolulu , Hawai‘ i , United States of America
Kaburu Stefano
Electronic publication date: 2025 May 22
Publication date: 2025
Volume: 13
Electronic Location ID: e19373
Received 2024 Nov 20; Accepted 2025 Apr 4
Copyright: ©2025 Santamaria et al.
Copyright year: 2025
Copyright holder: Santamaria et al.
License: This is an open access article, free of all copyright, made available under the Creative Commons Public Domain Dedication. This work may be freely reproduced, distributed, transmitted, modified, built upon, or otherwise used by anyone for any lawful purpose.
License URL: https://creativecommons.org/publicdomain/zero/1.0/

Keywords: Oniscidea, Cryptic species, Ligiidae, Intertidal, Species description, Pacific biodiversity

Funding: 2023–2024 University of Tampa Research Innovation and Scholarly Excellence Award GR3367 The molecular work completed as part of this study was funded by a 2023–2024 University of Tampa Research Innovation and Scholarly Excellence Award (RISE Grant GR3367). The funders had no role in study design, data collection and analysis, decision to publish, or preparation of the manuscript.

==============================
Isopods in the genus Ligia have been shown to harbor deeply divergent genetic lineages that have, in some instances, been recognized as cryptic species. For instance, the use of molecular taxonomic approaches to characterize coastal Ligia from the Hawaiian Islands led to the redescription of Ligia hawaiensis, the sole endemic coastal species previously recognized in the region, and to the description of seven new species endemic to the region. These species appear to be highly restricted to rift zones within single islands, single islands, or previously connected islands, suggesting these species evolved in allopatry. These findings, coupled with the poor dispersal capabilities exhibited by Ligia isopods and the geology of the Hawaiian Islands, suggest that additional cryptic species may exist in highly isolated populations yet to be studied. Studies to date have characterized Ligia from throughout the younger Hawaiian Islands (e.g., Kaua‘ i, O‘ ahu, Moloka‘ i, Maui, Lanai, and Hawai‘ i); however, no endemic Ligia populations from the older islands and more remote islands that form part of the Papahānaumokuākea Marine National Monument (PMNM) have been studied. This region represents the largest marine conservation area in the USA, and includes at least three islands where L. hawaiensis have been previously reported from. Herein, we apply molecular taxonomic approaches to characterize Ligia specimens from Nihoa, a remote island in the PMNM. Results show that Ligia from Nihoa form a highly divergent that is reciprocally monophyletic lineage with other Hawaiian Ligia species. This lineage, described as Ligia barack sp. nov., adds to the known biodiversity of the PMNM and highlights the importance of continued exploration and conservation of this remote and highly biodiverse region.

Introduction

The Hawaiian Islands (hereafter HI) are a series of islands, atolls, islets, and rocky outcroppings of volcanic origin spanning ∼2,400-km of the northern Pacific Ocean. Islands in the archipelago are arranged in a relatively linear manner, with younger islands located towards the eastern end of the archipelago and older islands found in its western end. Younger islands include eight major islands, all of which have formed in the past 5 million years (in decreasing age: Ni‘ ihau, Kaua‘ i, O‘ ahu, Moloka‘ i, Maui, Lanai, Kaho‘ olawe, and Hawai‘ i). The older islands, found west of Kaua‘ i, include ten island groups ranging widely in size and elevation (in decreasing age: Kure Atoll, Midway Atoll, Pearl & Hermes Atoll, Lisianski, Laysan, Maro Reef, Gardner Pinnacles, French Frigate Shoals, Necker, and Nihoa). These islands are part of the Papahānaumokuākea Marine National Monument (hereafter “PMNM”), a protected area of the United States of America established by presidential decree on June 15, 2006 to protect natural and cultural resources from the region. The Monument initially protected 362,073 km2 of marine habitats; however, it was extended by President Barack H. Obama in 2016 to encompass 1,508,870 km2 of the Pacific Ocean. This makes the PMNM the largest contiguous fully protected conservation in the United States of America and one of the largest marine preserves in the world. The habitats of the PMNM support an incredible diversity of coral, fish, birds, marine mammals and other flora and fauna, many of which are endemic to the PMNM (Starr & Martz, 1999; Starr & Starr, 2008; Kane, Kosaki & Wagner, 2014). Nonetheless, recent descriptions of new species from the PMNM suggest additional new species may exist in this region (Stein & Drazen, 2014; Pyle, Greene & Kosaki, 2016; Sherwood et al., 2020; Alvarado et al., 2022; Sherwood et al., 2022).

Intertidal habitats of the PMNM are known to harbor Ligia isopods, a genus of poorly dispersing isopods shown to hold high levels of cryptic diversity (Taiti et al., 2003; Hurtado, Mateos & Santamaria, 2010; Eberl et al., 2013; Santamaria et al., 2013; Raupach et al., 2014; Santamaria, Mateos & Hurtado, 2014; Santamaria et al., 2017; Greenan, Griffiths & Santamaria, 2018; Santamaria, 2019). Currently, nine Ligia species are thought to be endemic to the HI: eight coastal species that inhabit rocky intertidal habitats, and a terrestrial species that inhabit terrestrial habitats at elevation in the islands of Kaua‘ i, O‘ ahu and Hawai‘ i. The eight coastal species were formerly recognized as L. hawaiensis Dana 1853; however, they were split into these species on the basis of molecular, morphological, and geographic distributional data (Santamaria, 2019). Despite reports of “L. hawaiensis” from the islands of Nihoa, Necker, and La Perouse Pinnacle in the PMNM (Taiti & Howarth, 1996), which are located 190 to 675-km from Kaua‘ i, no specimens from these islands were included in any of the molecular studies characterizing Ligia from the HI to date (Taiti et al., 2003; Santamaria et al., 2013; Santamaria, 2019). Given the limited dispersal potential exhibited of Ligia isopods and the long-term isolation of these oceanic islands, molecular characterizations of these populations are likely to uncover additional cryptic species of Ligia in the region.

In this study, we use molecular approaches to characterize Ligia isopods from the island of Nihoa, the easternmost island in the PMNM. Doing so, we aim to determine: (a) whether Ligia individuals from this highly remote island harbor any unique genetic lineages, (b) if so, what are the phylogenetic relationships of these lineages to other Ligia species previously reported from the HI, (c) whether these lineages are divergent enough to be considered a novel species, and if so (d) describe said lineages as a new species. We do so by incorporating phylogenetic reconstructions, and distance- and phylogeny-based molecular species delimitation methods on a multi-locus dataset comprised of all extant Ligia species from the Hawaiian Islands and newly collected specimens from Nihoa. Our results indicate Ligia from Nihoa represent a highly divergent genetic lineage that is reciprocally monophyletic with all other Ligia species from the HI. Given its genetic uniqueness and geographic isolation, we describe this lineage as Ligia barack sp. nov. on the basis of molecular characters. The formal description of this cryptic species adds to our understanding of the biodiversity of the PMNM.

Materials and Methods

Sample collection

Ligia specimens were collected from the splash zone of rocky coastlines of Hanaka‘ ie‘ ie (Adam’s Bay) in Nihoa during April of 2023. All individuals were caught by hand and field-preserved in 70% isopropanol. The collection of specimens from Nihoa was conducted under a permit granted to the Papahanaumokuakea Marine National Monument Co-Trustees, which include the US Fish and Wildlife Service, by the State of Hawai‘ i Board of Land and Natural Resources (Permit Number PMNM-2022-001). Once in the laboratory, specimens were transferred to 70% ethanol. We identified male individuals as members of the L. hawaiensis cryptic species complex by inspecting the morphology of the distal process of the endopod of the 2nd pleopod and comparing to previous reports (Taiti et al., 2003). Specimens from Nihoa were then inspected under an AmScope SM-4TZ-144 3.5X-90X Trinocular Zoom Stereo Microscope equipped with a 20 MP imaging system. Drawings of characters were made electronically from images produced in the above mentioned system.

Molecular laboratory methods

We used Zymo Research’s Quick g-DNA MiniPrep Kit to extract total genomic DNA for six Ligia individuals collected in Nihoa. DNA was extracted from 2–3 pereopods per individual using standard protocol instructions. We then used previously published primers and conditions to polymerase chain reaction (PCR) amplify the same four mitochondrial and three nuclear gene fragments used by Santamaria (2019) to conduct a taxonomic revision of L. hawaiensis: (a) a 658-bp segment of the Cytochrome Oxidase I gene using primers LCO-1490/HCO-2198 (hereafter COI, primers LCO1490/HCO2198; Folmer et al., 1994), (b) a ∼490-bp segment of the 16S rRNA gene using primers 16Sar/16Sbr (primers 16Sar/16Sbr; Palumbi, 1996), (c) a ∼495-bp segment of the 12S rDNA gene using primers crust-12Sf/crust-12Sr (primers crust-12Sf/crust-12Sr; Podsiadlowski & Bartolomaeus, 2005), (d) a 361-bp fragment of the Cytochrome-b gene using primers 144F and 270R to amplify (hereafter Cyt-b, primers 144F/151F and 270R/272R; Merritt et al., 1998), (e) a ∼1,000-bp segment of the 28S rDNA gene using primers 28SA/28SB (primers 28SA/28SB Whiting, 2002), (f) a 664-bp region of the alpha-subunit of the Sodium Potassium ATPase using primers NaK-forb/NaK-rev2 (hereafter NaK, primers NaK-forb/NaK-rev2; Tsang et al., 2008), and (g) a ∼328-bp fragment of the Histone H3 gene using primers H3AF/H3ARto amplify (primers H3AF/H3AR; Colgan et al., 1998). PCR products were visualized on 1% agarose gels, stained using Apex Safe DNA Gel Stain (Apex Bioresearch Products). Positive amplicons were sequenced at the Arizona Genetics Core.

Sequence alignment and model testing

Sequences were assembled, edited (i.e., had primers removed), and inspected for evidence indicative of heteroplasmy and/or heterozygosity (e.g., multiple peaks in chromatograms) in CodonCode Aligner v10.0.1. No evidence of heteroplasmy or heterozygosity was observed. Sequences produced in this study were then aligned and added to the aligned dataset produced by Santamaria in 2019 using the “—add” option of the MAFFT webserver (Katoh & Standley, 2013) using standard settings. This dataset includes all currently valid species of Ligia endemic to the Hawaiian Islands. Alignments for the three ribosomal genes included in this study (i.e., 28S rDNA, 16S rDNA, and 12S rDNA) were compared to those produced by Santamaria (2019) with poorly aligned sites removed. We inspected alignments of protein coding genes (i.e., COI, Cyt-b, NaK, H3A) and did not observe any evidence suggestive of pseudo-genes such as the presence of early stop codons or indels.

For each gene alignment, we selected the most appropriate model of nucleotide evolution from all available models in jModeltest v2.1 (Darriba et al., 2012) by evaluating their likelihood using a fixed BioNJ-JC tree under the Bayesian information criterion (BIC). Gene alignments were then concatenated using SequenceMatrix v.1.9 (Vaidya, Lohman & Meier, 2011). We used a similar approach as described above to select the most appropriate model of nucleotide evolution for the concatenated alignment. We also selected the most appropriate partition scheme to use in our phylogenetic reconstructions in PartitionFinder v2.1.1 (Lanfear et al., 2016) by evaluating different partitioning combinations of an a priori partitioning scheme that consisted of each ribosomal gene as a single partition with protein coding genes separated by gene and codon position. Partitioning schemes were evaluated under the BIC criterion and the following parameters: branch lengths = unlinked; models = all; model selection = BIC; search = greedy. Lastly, we estimated pairwise Kimura-2-parameter (K2P) distances in MEGA v11.0.13 (Kumar, Stecher & Tamura, 2016) for the COI dataset.

Phylogenetic reconstructions

We conducted phylogenetic reconstructions on the concatenated alignment of all gene fragments under both maximum likelihood and Bayesian inference approaches using two different partitioning approaches: by gene, and as determined by PartitionFinder. Maximum likelihood (ML) searches were conducted in RAxML-NG v1.1.0 (Kozlov et al., 2019) and consisted of 1,000 bootstrap replicates followed by a thorough ML search under the GTR +Γ model run with all other settings as default. Bayesian searches were conducted in MrBayes v3.2.7 (Ronquist & Huelsenbeck, 2003) and consisted of four separate runs consisting each of two chains, run for 20 × 106 generations, sampled every 1,000th generation. All other settings were as default. Bayesian searches were monitored to determine if they had reached and maintained stationarity using the following criteria: (a) stable posterior probability values; (b) high correlation between the split frequencies of independent runs as implemented in AWTY (Nylander et al., 2007); (c) small and stable average standard deviation of the split frequencies of independent runs; (d) potential scale reduction factor close to 1; and (e) an effective sample size (ESS) >200 for the posterior probabilities, as evaluated in Tracer v1.7.2 (Rambaut et al., 2018). For all searches, we calculated majority-rule consensus trees using the SumTrees command of DendroPy v3.10.1 (Sukumaran & Holder, 2010). For Bayesian analyses, samples prior to stationarity were discarded as burn-in.

Molecular species delimitation analyses (MSDAs)

We implemented both tree- and distance-based species delimitation analyses to determine whether our molecular dataset supports the identification of Ligia from Nihoa as a separate species. Tree-based molecular species delimination analyses (MSDAs) were carried out using the Poisson Tree Processes model as implemented in the PTP server (http://species.h-its.org/) and the General Mixed Yule Coalescent model (hereafter GMYC; Fujisawa & Barraclough, 2013). PTP analyses were carried out on all phylogenetic trees produced in RAxML and MrBayes. Settings used were as follows: 500,000 Markov chain Monte Carlo (MCMC) iterations; a burn-in of 0.10; and a thinning value of 100. As GMYC delineations require ultrametric trees as input, we estimated ultrametric trees for the unpartitioned concatenated mitochondrial dataset using BEAST v2.1.3 (Bouckaert et al., 2014) assuming a constant rate of evolution and speciation assuming a Yule process (i.e., constant speciation rate; Yule, 1925; Gernhard, 2008), and under a coalescent model of speciation assuming a constant population size (Kingman, 1982). Both searches were carried out for 50 million generations sampled every 1,000th generation using the most appropriate model of nucleotide evolution. Resulting trees were summarized using the SumTrees command with burn-in discarded and with edges set as per the mean-age option. Resulting ultrametric trees were analyzed using the GMYC approach as implemented by the ‘splits’ package (http://r-forge.r-project.org/projects/splits/) in R using default settings.

We conducted distance-based analyses using ASAP (Puillandre et al., 2012) on the COI gene dataset alone after masking ambiguous sites using the ASAP webserver (https://bioinfo.mnhn.fr/abi/public/asap/). ASAP analyses were carried out under the Kimura 2-parameter (K2P) nucleotide evolution model, with all other settings as default. We used KoT (Spöri et al., 2022) to estimate the K/θ ratio (Birky Jr et al., 2010; Birky Jr, 2013) between Ligia from Nihoa and their most closely related taxa identified by phylogenetic analyses. Analyses were carried on the concatenated dataset assuming a K/θ threshold of 4, a value that represents a >95% probability that sister clades have become reciprocally monophyletic (Birky Jr, 2013).

We evaluated the following criteria to determine whether Ligia from Nihoa represent a novel species in need of description: (1) did all phylogenetic reconstructions place all Nihoa Ligia individuals in a well-supported (bootstrap support (BS) >90%, Bayesian posterior probability (BPP) >95%) monophyletic clade that excluded all other Ligia from the Hawaiian Islands; (2) were pairwise COI K2P distances amongst Ligia Nihoa specimens <1.0%; (3) do comparisons between Ligia from Nihoa and its sister taxon produce a K/θ>4 (i.e., 4X rule; Birky Jr, 2013); (4) did most MSDAs separate Nihoa individuals as a putative species; (5) did this putative species exclude all other Ligia from the HI. As the answer for all these criteria was affirmative, we herein describe Ligia barack, a novel species of Ligia from Nihoa. We determined diagnostic nucleotide positions for this novel species using FASTACHAR v0.2.4 (Merckelbach & Borges, 2020) by comparing L. barack sp. nov. to all other Ligia species endemic to the HI included in the dataset used in this study.

The electronic version of this article in Portable Document Format (PDF) will represent a published work according to the International Commission on Zoological Nomenclature (ICZN), and hence the new names contained in the electronic version are effectively published under that Code from the electronic edition alone. This published work and the nomenclatural acts it contains have been registered in ZooBank, the online registration system for the ICZN. The ZooBank LSIDs (Life Science Identifiers) can be resolved and the associated information viewed through any standard web browser by appending the LSID to the prefix http://zoobank.org/. The LSID for this publication is: urn:lsid:zoobank.org:pub:6CE79D26-19BA-435D-94A8-5A822ADD42B0. The online version of this work is archived and available from the following digital repositories: PeerJ, PubMed Central and CLOCKSS.

Results

We successfully produced sequences for four mitochondrial and three nuclear genes for six Ligia specimens from Nihoa (hereafter L. barack sp. nov). Unique haplotypes have been deposited in GenBank and BOLD (see Table 1 for GenBank Accession Numbers). The addition of these sequences to the alignment produced by Santamaria (2019) produced a concatenated dataset 3,996-bp long prior to the removal of poorly aligned positions for the 16S, 12S, and 28S rDNA gene. This final alignment included 196 individuals from 40 localities in the HI including Nihoa (Fig. 1, Table 1). Removal of poorly aligned sites (43, 17, and 49 for the 16S, 12S, and 28S rDNA genes respectively) produced a final alignment 3,887-bp long containing 543 parsimony informative sites (COI = 185; Cyt-b: 120; 12S rDNA = 99; 16S rDNA = 91; 28S rDNA = 39; NaK = 6; H3A = 3). An annotated alignment is provided as Dataset S1.

Table 1 Localities included in the study, with corresponding number of individuals included, GenBank accession numbers, and geographic information.

Loc. Label	Locality
Name	New
Loc.	# inds.	COI
Acc. No	16S rDNA
Acc, No.	12S rDNA.
Acc. No.	Cytb
Acc. No.	28S rDNA
Acc. No.	NaK
Acc. No.	H3A
Acc. No.	Latitude	Longitude	
A1	Wai‘Ōpae
Maui	NO	2	MK034488	MK032502
KF546549	MK032601
KF546573	MK034572
KF546718	N/A	N/A	MK034658	20°37′29.20″N	156°12′34.10″W	
A2	Kealakukea Bay
Hawai‘ i	NO	6	MK034474
MK034475
MK034476
MK034477
KF546627	MK032515
MK032516
MK032517
MK032518
MK032519	MK032608
MK032609
KF546574		MK940873
MK940874	KF546594	MK034663	19°28′32.88″N	155°55′11.04″W	
A3	Pu‘ unalu Beach Park
Hawai‘ i	NO	5	MK034513
MK034514
KF546628	MK032564
MK032565
MK032566
MK032567
KF546551	MK032627
MK032628
KF546576	MK034582
MK034583
KF546716	MK940887
KF546701	KF546593	MK034677	19°08′00.60″N	155°30′18.30″W	
A4	Isaac Hale Beach Park
Hawai‘ i	NO	6	N/A	MK032568
MK032569
MK032570
MK032571
MK032572
KF546550	MK032629
MK032630
KF546575	MK034584
MK034585
KF546717	MK940888
KF546702	MK034645
MK034646
KF546586	MK034678
MK034679	19°27′26.82″N	154°50′31.68″W	
A5	Miloli Beach Park
Hawai‘ i	NO	5	MK034478
MK034479
MK034480
MK034481
MK034482	MK032554
MK032555
MK032556
MK032557
MK032558	MK032623
MK032624	MK034567
MK034568
MK034569	MK940885
MK940886	MK034642
MK034643	MK034675	19°10′58.10″N	155°54′25.10″W	
A6	Waianapanapa
State Park Maui	NO	5	N/A	MK032492
MK032493
MK032494
MK032495
MK032496	MK032596
MK032597	MK034570
MK034571	MK940866	MK034605	MK034654
MK034655	20°47′21.80″N	156°00′07.90″W	
A7	Koki Beach Park
Maui	NO	5	MK034483
MK034484
MK034485
MK034486
MK034487	MK032497
MK032498
MK032499
MK032500
MK032501	MK032598
MK032599
MK032600		MK940867	MK034606
MK034607
MK034608
MK034609
MK034610	MK034656
MK034657	20°43′41.62″N	155°59′06.71″W	
B1	Nu‘ uanu Pali
O‘ ahu	NO	1	KF546661	KF546548	KF546572	KF546719	N/A	N/A	N/A	N/A	N/A	
C1	Mt Kahili
Kaua‘ i	NO	1	KF546660	KF546546	KF546578	N/A	N/A	N/A	N/A	N/A	N/A	
C2	Makaleha Mts
Kaua‘ i	NO	1	KF546659	KF546545	KF546577	KF546723	N/A	N/A	N/A	N/A	N/A	
C3	Haupu Range
Kaua‘ i	NO	1	KF546655	KF546547	KF546579	KF546722	KF546683	KF546592	N/A	N/A	N/A	
D1	Kalihiwai Beach
Kaua‘ i	NO	14	MK034540
MK034541
MK034542
MK034543
MK034544
KF546598
KF546599
KF546600
KF546601
KF546602
KF546603
KF546604
KF546605
KF546606	MK032544
MK032545
MK032546
MK032547
MK032548
KF546544	MK032619
MK032620
KF546571	MK034593
MK034594
KF546721	MK940882
MK940883
KF546686
KF546687
KF546688
KF546689
KF546690	MK034635
MK034636
MK034637
MK034638
MK034639
KF546585	MK034672
MK034673	22°13′05.30″N	159°25′31.15″W	
D2	Kauapea Beach
Kaua‘ i	NO	1	KF546656	KF546543	KF546570	KF546720	N/A	N/A	N/A	N/A	N/A	
D6	Hoai Bay
Kaua‘ i	NO	5	MK034545
MK034546
MK034547
MK034548
MK034549	MK032549
MK032550
MK032551
MK032552
MK032553	MK032621
MK032622	MK034595
MK034596	MK940884	MK034640
MK034641	MK034674	21°52′51.93″N	159°28′25.01″W	
D7	Hanaka‘ ie‘ ie (Adam’s Bay), Nihoa	YES	6	PP851829
PP851830
PP851831
PP851832
PP851833
PP851834	PP852382
PP852383
PP852384
PP852385
PP852386
PP852387	PP852388
PP852389
PP852390
PP852391
PP852392
PP852393	PP856001
PP856002
PP856003
PP856004
PP856005
PP856006	PP852394
PP852395
PP852396
PP852397
PP852398
PP852399	PP856007
PP856008
PP856009
PP856010
PP856011
PP856012	PP861092
PP861093
PP861094
PP861095
PP861096
PP861097	23°03′30.30″N	161°55′27.60″W	
E2	Papohaku Beach Park
Moloka‘i	NO	1	KF546607	KF546542	KF546569	KF546715	N/A	N/A	N/A	21°10′46.56″N	157°15′5.88″W	
E3	North of Puko‘ o
Lana‘i	NO	9	KF546608
KF546609
KF546610
KF546611
KF546612
KF546613
KF546614
KF546615
KF546616	KF546540	KF546565	KF546713	KF546696
KF546697
KF546698
KF546700	KF546587	N/A	21°06′06.84″N	156°45′06.66″W	
E4	Manele Bay
Moloka‘i	NO	7	KF546643
KF546644
KF546645
KF546646
KF546647
KF546648
KF546649	KF546538	KF546564	N/A	KF546677
KF546678
KF546679
KF546680
KF546681
KF546682	KF546589	N/A	20°44′37.37″N	156°53′12.47″W	
E5	Poelua Bay
Maui	NO	1	KF546657	KF546532	KF546566	KF546710	N/A	N/A	N/A	N/A	N/A	
E6	Spreckelsville
Maui	NO	8	KF546595
KF546596
KF546597
KF546650
KF546651
KF546652
KF546653
KF546654	KF546539	KF546567	KF546712	KF546691
KF546692
KF546693
KF546694
KF546695	KF546590	N/A	20°54′31.38″N	156°24′40.26″W	
E7	Keanae
Maui	NO	6	KF546658	MK032487
MK032488
MK032489
MK032490
MK032491
KF546537	MK032594
MK032595
KF546568	MK034597
MK034598
KF546714	MK940865	N/A	MK034652
MK034653	N/A	N/A	
E8	DT Fleming Beach Park
Maui	NO	2	MK034550
MK034551	MK032503
MK032504	MK032602
MK032603	MK034599
MK034600	MK940868
MK940869	MK034611
MK034612	MK034659
MK034660	21°00′20.82″N	156°38′58.43″W	
E9	Hanakao‘ o Park
Maui	NO	5	MK034552
MK034553
MK034554
MK034555
MK034556	MK032505
MK032506
MK032507
MK032508
MK032509	MK032604
MK032605	MK034601
MK034602	MK940870
MK940871	MK034613
MK034614
MK034615
MK034616	N/A	20°54′34.10″N	156°41′19.03″W	
E10	Wawamalu Beach Park
O‘ ahu	NO	5	MK034557
MK034558
MK034559
MK034560
MK034561	MK032535
MK032536
MK032537
MK032538
MK032534	MK032616
MK032617	MK034603
MK034604	MK940879	MK034628
MK034629
MK034630
MK034631
MK034632	MK034669	21°17′12.51″N	157°40′07.66″W	
F1	Pupukea
O‘ ahu	NO	16	MK034494
MK034495
MK034496
MK034497
KF546617
KF546618
KF546619
KF546620
KF546621
KF546622
KF546623
KF546624
KF546625
KF546626	MK032520
MK032521
MK032522
MK032523
KF546533
KF546531	MK032610
MK032611
KF546562	MK034575 MK034591
KF546709	KF546667
KF546668
KF546669
KF546670
KF546671	MK034621
MK034622
MK034623
KF546591	MK034664
MK034665	21°38′59.70″N	158°03′45.48″W	
F2	Pouhala Marsh
O‘ ahu	NO	1	N/A	KF546532	N/A	KF546710	N/A	N/A	N/A	N/A	N/A	
F3	Honomanu Bay
Maui	NO	1	N/A	KF546530	KF546563	KF546708	N/A	N/A	N/A	N/A	N/A	
F4	Keokea Beach
Hawai‘ i	NO	1	N/A	KF546529	KF546558	KF546703	N/A	N/A	N/A	N/A	N/A	
F5	Onekahakaha Beach Park
Hawai‘ i	NO	19	MK034520
MK034521
MK034522
MK034523
MK034524
KF546629
KF546630
KF546631
KF546632
KF546633
KF546634
KF546635
KF546636
KF546637
KF546638
KF546639
KF546640
KF546641
KF546642	MK032573
MK032574
MK032575
MK032576
KF546534	MK032631
MK032632
KF546561	MK034588
KF546705	KF546672
KF546673
KF546674
KF546675
KF546676	KF546588	MK034680
MK034681	19°44′16.05″N	155°02′20.15″W	
F6	Leleiwi Beach
Hawai‘ i	NO	1		KF546535	KF546560	KF546706	N/A	N/A	N/A	N/A	N/A	
F7	South Point
Hawai‘ i	NO	6	MK034515
MK034516
MK034517
MK034518
MK034519	MK032559
MK032560
MK032561
MK032562
MK032563
KF546536	MK032625
MK032626
KF546559	MK034586
MK034587
KF546707	N/A	MK034644	MK034676			
F8	Kapa‘ a State Park
Hawai‘ i	NO	1		KF546528	KF546557	KF546704	N/A	N/A	N/A	20°12′11.52″N	155°54′6.66″W	
F9	Kolekole Beach Park
Hawai‘ i	NO	5	MK034525
MK034526
MK034527
MK034528
MK034529	MK032577
MK032578
MK032579
MK032580
MK032581	MK032633
MK032634	MK034589
MK034590	MK940891
MK940892	MK034647	N/A	19°52′58.80″N	155°07′07.60″W	
F10	Laupahoehoe Beach Park
Hawai‘ i	NO	5	MK034530
MK034531
MK034532
MK034533
MK034534	MK032582
MK032583
MK032584	MK032635
MK032636	MK034591	MK940893
MK940894
MK940895
MK940896	MK034648	MK034682
MK034683	19°59′36.60″N	155°14′24.01″W	
F11	Spencer Beach Park
Hawai‘ i	NO	5	MK034535
MK034536
MK034537
MK034538
MK034539	MK032585
MK032586
MK032587
MK032588
MK032589	MK032637
MK032638	MK034592	N/A	MK034649
MK034650
MK034651	MK034684
MK034685	20°01′22.41″N	155°49′21.50″W	
F12	Baby Beach
Maui	NO	7	MK034562
MK034563
MK034564
MK034565
MK034566	MK032482
MK032483
MK032484
MK032485
MK032486	MK032592
MK032593	N/A	MK940864	N/A	N/A	20°54′45.09″N	156°24′16.01″W	
F13	Kahaluu
O‘ ahu	NO	5	MK034489
MK034490
MK034491
MK034492
MK034493	MK032510
MK032511
MK032512
MK032513
MK032514	MK032606
MK032607	MK034573
MK034574	MK940872	MK034617
MK034618
MK034619
MK034620	MK034661
MK034662	21°28′17.81″N	157°50′40.65″W	
F14	Kaena Point (North)
O‘ ahu	NO	5	MK034498
MK034499
MK034500
MK034501
MK034502	MK032524
MK032525
MK032526
MK032527
MK032528	MK032612
MK032613	MK034576
MK034577	MK940875
MK940876	MK034624
MK034625	MK034666
MK034667	21°34′47.46″N	158°14′15.43″W	
F15	Kaiaka Bay Beach Park
O‘ ahu	NO	5	MK034503
MK034504
MK034505
MK034506
MK034507	MK032529
MK032530
MK032531
MK032532
MK032533	MK032614
MK032615	MK034578
MK034579	MK940877
MK940878	MK034626
MK034627	MK034668	21°35′20.62″N	158°07′03.42″W	
F16	Kaena Point (South)
O‘ ahu	NO	5	MK034508
MK034509
MK034510
MK034511
MK034512	MK032590
MK032539
MK032540
MK032541
MK032542
MK032543	MK032618	MK034580
MK034581	MK940880
MK940881	MK034633
MK034634	MK034670
MK034671	21°33′21.21″N	158°14′54.88″W	

Figure 1 Ligia localities included in this study.

Labels and colors correspond with other figures and tables in this study and that of Santamaria et al. (2013) and Santamaria (2019). Detailed information for each locality is presented in Table 1. Localities of the suppralittoral Ligia included: Kaua‘ i: D1-Kalihiwai Beach, D2-Kauapea Beach, D6-Hoai Bay; D7- Hanaka‘ ie‘ ie (Adam’s Bay); O‘ ahu: E10-Wawamalu Beach Park, F1-Pupukea, F2-Pouhala Marsh, F13-Kahaluu, F14-Kaena Point (North), F15-Kaiaka Bay Beach Park, F16-Kaena Point (South); Moloka‘ i: E2-Papohaku Beach Park, E4-Manele Bay; Lana‘ i: E3-North of Puko‘ o; Maui: A1-Wai‘ Ōpae; A6-Waianapanapa State Park, A7-Koki Beach Park, E5-Poelua Bay, E6-Spreckelsville, E7-Keanae, E8-DT Fleming Beach Park, E9-Hanakao‘ o Park, F3-Honomanu Bay, F12-Baby Beach Spreckelsville Area; Hawai‘ i: A2-Kealakukea Bay, A3-Pu‘ unalu Beach Park, A4-Isaac Hale Beach Park, A5-Miloli Beach Park, F4-Keokea Beach, F5-Onekahakaha Beach Park, F6-Leleiwi Beach, F7-South Point, F8-Kapa‘ a State Park, F9-Kolekole Beach Park, F10-Laupahoehoe Beach Park, F11-Spencer Beach Park. Localities of the terrestrial L. perkinsi included are Kaua‘ i: C1-Mt Kahili, C2-Makaleha Mts, C3-Haupu Range; O‘ ahu: B1-Nu‘ uanu Pali. World map is edited from a public domain map produced by Colohisto. Original vector map is available at https://commons.wikimedia.org/wiki/File:BlankMap-World_1990.svg. Map of the Hawaiian Islands is reproduced from Santamaria (2019). Map is available at: https://doi.org/10.7717/peerj.7531/fig-1.

All phylogenetic reconstructions completed in this study were similar to those reported by Santamaria et al. (2013) and Santamaria (2019), with the exception of L. barack sp. nov individuals (Fig. 2). These individuals were placed in a well-supported clade (BS = 100; PP = 100, Fig. 2) that excluded individuals from all other Ligia species endemic to the HI. Our phylogenetic reconstructions identified four highly divergent and reciprocally monophyletic lineages consisting solely of coastal Ligia endemic to the Hawaiian Islands: (a) Clade A (lavenders and purples in all figures; BS = 100; posterior probability (PP) = 100) which consisted of all L. dante (A2, A5 in Hawai‘ i), L. honu (A3–4 in Hawai‘ i) and L. eleluensis (A1, A6–7 in Maui) individuals; (b) Clade D (green in all figures; BS =100; PP =100) which included all L. hawaiensis individuals from Kaua‘ i (D1–2, D6) as well as L. barack sp. nov. (D7); (c) Clade E (oranges and yellows in all figures; BS = 100; PP = 100) consisting of all L. mauinuiensis individuals from O‘ ahu (E10), Moloka‘ i (E2, E3), Lana‘ i (E4), and Maui (E5–E9); and lastly (d) Clade F (reds in all figures; BS = 100; PP = 100) which included all L. kamehameha (F4–F11 in Hawai‘ i), L. rolliensis (F1–2 and F13–16 in O‘ ahu), and L. pele (F3, F12 in Maui) individuals. We also observed two lineages consisting of individuals of the terrestrial L. perkinsi: (a) Clade B (black in all figures) from O‘ ahu (B1), and (b) Clade C (blue in all figures; BS = 100; PP = 100) from Kaua‘ i (C1–3).

Figure 2 Majority rule consensus tree of bootstrap replicates produced by analyzing the concatenated mitochondrial and nuclear dataset of Ligia from the Hawaiian Islands in RAxML under the GTR +Γ under a “by gene” partitioning scheme.

Branches and clades are colored as per Santamaria et al. (2013) and Santamaria (2019). Values by nodes correspond with bootstrap support values observed in RAxML analyses (above) and posterior probabilities produced in MrBayes analyses (below). Asterisks (*) denote 100% support across all analyses.

Clades D, E, and F were placed in a well-supported monophyletic group (BS = 100; PP = 100) with clades E and F identified as each other’s sister clade (BS = 81–100; PP = 91–97). The sister to the “D + E + F” clade was Clade C (BS = 99–100; PP = 100), which consisted of the terrestrial L. perkinsi from Kaua‘ i. Clade B, consisting of the terrestrial L. perkinsi from O‘ ahu, was identified as the sister clade to the large monophyletic group consisting of clades C, D, E, and F (BS = 100; PP = 100). The most basal group was Clade A, which consisted of coastal Ligia species from the islands of Maui and Hawai‘ i.

COI K2P distances between Ligia species from HI ranged between 3.0–17.8%, with comparisons between L. barack sp. nov and other species in the region ranging between 3.0–17.8% (Table 2). Genetic diversities produced when comparing L. barack sp. nov were low, ranging between 0.0–0.3% (Table 2).

Table 2 Estimates of evolutionary divergence, as measured by Kimura 2-parameter distances, for Ligia barack and other Ligia species from the Hawaiian Islands.

	L. barack	L. dante	L. honu	L. eleluensis	L. perkinsi
(O‘ ahu)	L. perkinsi
(Kaua‘ i)	L. hawaiiensis	L. mauinuiensis	L. rolliensis	L. kamehameha	
L. barack	0.0–0.3
(0.2)										
L. dante	13.9–15.2
(14.5)	0.0-4.6
(2.4)									
L. honu	14.9–15.1
(15)	5.8–7.5
(6.8)	N/A								
L. eleluensis	17–17.8
(17.4)	9.7–11.2
(10.7)	10.9–11.3
(11.0)	0.0–0.9
(0.5)							
L. perkinsi
(O‘ ahu)	14.8–15.0
(14.9)	14.2–15.0
(14.5)	15.1–15.1
(15.1)	14.3–14.8
(14.6)	N/A						
L. perkinsi
(Kaua‘ i)	15.8–16.9
(16.4)	11.9–14.3
(13.2)	13.7–15.1
(14.1)	12.5–13.9
(13.2)	14.5–15.3
(14.9)	1.0–2.7
(1.9)					
L. hawaiiensis	3.0–4.0
(3.5)	12.8–15.4
(14.4)	15.0–16.9
(16.4)	14.7–16.4
(15.8)	14.8–15.8
(15.5)	13.8–15.6
(15.1)	0.0–2.2
(0.9)				
L. mauinuiensis	11.5–13.2
(12.5)	13.6–15.9
(14.9)	14.5–15.3
(15.0)	13.9–16.4
(15.3)	16.0–16.6
(16.4)	12.5–14.2
(13.0)	10.3–12.7
(11.5)	0.0–2.4
(0.7)			
L. rolliensis	12.5–14.3
(13.6)	14.0–16.6
(15.3)	14.9–16.2
(15.3)	13.4–16.2
(14.5)	13.7–14.7
(14.1)	12.9–16.1
(13.9)	12.5–14.8
(13.6)	10.9–12.8
(11.6)	0.0–7.1
(1.9)		
L. kamehameha	12.3–15.4
(13.5)	13.8–16.4
(14.8)	13.6–15.5
(14.6)	13.8–15.4
(14.6)	12.6–13.9
(13.2)	11.1–15.0
(13.2)	11.6–14.6
(12.9)	8.7–13.1
(10.6)	4.0–8.7
(5.3)	0.0–5.4
(2.5)	

Molecular species delimitations consistently identified L. barack sp. nov as a separate and distinct species from other Ligia species endemic to the HI. ASAP analyses of the COI dataset placed all Nihoa specimens in a separate putative species containing no Ligia from other localities in nine of the ten best partitions produced by ASAP, with only the ninth best supported partition (p-value rank = 4; W rank = 18; threshold distance = 0.040249) grouping Nihoa Ligia with L. hawaiensis individuals (Kaua‘ i). All tree-based MSDAs carried out in PTP, bPTP, and GMYC recognized L. barack sp. nov as a separate species. Lastly, comparisons between this newly described species and L. hawaienesis, its sister taxon, in KoT produced a K/θ ratio of 9.912.

Taxonomy

Based on the long-term and geographical isolation for Nihoa, morphological comparisons, results of phylogenetic reconstructions and MSDAs, COI K2P pairwise distances reported herein, and K/θ ratio between it and its sister taxon, we describe Ligia barack sp. nov., a new species of Ligia from Nihoa. A holotype and three paratypes were deposited at the Florida Museum of Natural History (FLMNH) in Gainesville, FL, USA. We describe L. barack sp. nov. primarily using molecular characters and some diagnostic morphological characters. We include a broad description of the holotype that covers traits evaluated by previous authors (Taiti et al., 2003; Santamaria et al., 2013; Santamaria, 2019), a photograph of the holotype of L. barack (Fig. 3), and illustrations of diagnostic features of this species (Fig. 4). Other traits not mentioned below (e.g., pereopods 2–6) are as described and/or illustrated by Taiti et al. (2003), Taiti, Ferrara & Kwon (1992), and Jackson (1933).

Figure 3 Holotype of Ligia barack, a new species from Nihoa.

Holotype shown in this picture is deposited at the Florida Museum of Natural History (UFID 72496).

Figure 4 Ligia barack nov. sp. holotype (UFID 72496).

(A) Cephalon; (B) peduncles 1–3 of antenna; (C) Pleotelson, dorsal view; (D) Pereopod 1; (E) Pereopod 6; (F) Closeup of dactylus of Pereopod 6; (G) apex of appendix masculina of second pleopod.

Ligia barack nov. sp.

LSID: urn:lsid:zoobank.org:act:558494AB-37D7-47BA-BA54-4E532D7585C6.

BOLD BINs: AFQ9578.

Materials examined: six individuals from the island of Nihoa (D7). Both males and females were included. The holotype (♂; UFID 72496; Fig. 3), and three paratypes (♀♀♀; UFID 72497-72499) from the type locality have been deposited at the Florida Museum of Natural History (FLMNH) in Gainesville, FL, USA.

Type locality: Hanaka‘ ie‘ ie (Adam’s Bay), Nihoa, Hawai‘ i, USA. (D7; 23°03′30.3″N 161° 55′27.6″W).

Description of holotype: male individual that is 17.8 mm long and 6.7 mm wide at the widest point of the pereionite 4 (body length to width ratio of ∼2.7; Fig. 2). Eyes are large (eye length is ∼0.5 greatest width of cephalon) and closely spaced (inter-eye distance ∼0.5 times eye length) (Fig. 4A). Posterolateral processes of the pereionite 7 extend ∼13 length of the pleonite 3. Antennae extends just past midbody, being about ∼0.6 times the total body length. Article 3 of the peduncle of the antenna is about 1.5X length of the peduncle article 2, with robust setae on either side of the distal end (Fig. 4B). Pleotelson shape similar to that of other Ligia in the Hawaiian Islands; however, the lateral posterior points are nearly parallel to the interior lateral posterior points (Fig. 4C). First three pereopods have papillar fields present, with those in the carpus and merus of the first pereopod present in over half of pereopod segment (e.g., pereopod 1 illustrated in Fig. 4D). The dactylus of the sixth and seventh pereopod have four small robust setae on the tergal margin (Fig. 4F). Apex of the appendix masculina in the endopod of the second pleopod is enlarged and slightly bilobed with distal end rounded (Fig. 4G).

The holotype is deposited in the FLMNH under UFID 72496. GenBank Accession numbers for sequences obtained from the holotype are: PP851829 (COI); PP852382 (16S rDNA); PP852387 (12S rDNA); PP856001 (Cyt-b); PP852394; (28S rDNA); PP856007 (NaK); and PP861092 (H3A).

Remarks: the herein described L. barack sp. nov can be distinguished from other coastal Ligia species endemic to the Hawaiian Islands by the absence of a tuft of very long thin setae on the tergal margin of the dactylus of the sixth and seventh pereopods (Figs. 4E–4F; see Fig. 4 in Taiti et al., 2003). The appendix masculina of the endopod of the second pleopod also differs between L. barack sp. nov and other species in the region (Taiti et al., 2003). The appendix masculina of L. barack is slightly bilobed with the distal end being rounded, which contrasts with the obliquely truncate morphology seen in coastal Ligia from the Hawaiian Islands (see Fig. 4G; contrast with Fig. 6 in Taiti et al., 2003). Lastly, L. barack exhibits slight differences in eye shape and size. This species has large eyes (eye length is ∼0.5 greatest width of cephalon) which is similar to other coastal Ligia species endemic to the Hawaiian Islands; however, the eyes appear to be the most distantly spaced of all these species. The inter-eye distance of L. barack is about ∼0.5 times the eye length, while the smallest ratio for other nearby species is ∼0.7 (seen in L. honu, L. hawaiiensis, L. kamehameha, L, mauiniensis, L. rolliensis; Santamaria, 2019). Lastly L. barack can also be distinguished using molecular characters listed below.

Diagnostic molecular characters:

COI: 1-C; 31-A; 94-C; 526-C

16S: 288(316)-T.

Cyt-b: 181-G; 223-C; 262-G; 265-G; 354-G

12S: 380(398)-G.

Distribution: rocky intertidal habitats of Nihoa.

Hawaiian common name: Pokipoki o Hanaka‘ ie‘ ie. Pokipoki is the Hawaiian name for terrestrial isopods and similar creatures inhabiting aquatic and terrestrial habitats. Meanwhile, Hanaka‘ ie‘ ie refers to the traditional name for Adam’s Bay of Nihoa Island. Thus, this name broadly translates to “the isopod from Adam’s Bay of Nihoa Island.”

Etymology: this species is named after Barack H. Obama, the former President of the United States of America, who was born in the island of O‘ ahu and who is responsible for the expansion of the Papahānaumokuākea Marine National Monument to its current size.

Discussion

The Hawaiian Islands (HI) were previously thought to harbor a single endemic coastal Ligia species: Ligia hawaiensis. This species, first described by Dana (1853), was determined to represent a cryptic species complex composed of allopatric species with distributional ranges largely limited to rift zones within a single island, single islands, or previously connected islands (Santamaria, 2019). Despite previous reports of L. hawaiensis from the remote and older islands found in the Papahānaumokuākea Marine National Monument (Taiti & Howarth, 1996), none of the molecular studies conducted on Hawaiian Ligia to date have included populations from these islands. This has left unanswered whether Ligia populations from the older and highly remote islands in the PMNM harbor highly divergent genetic lineages and/or novel species in need of description. By using similar molecular approaches to those used by Santamaria (2019) to describe highly genetically divergent yet largely morphologically cryptic lineages of Ligia in the HI as new species, we herein describe L. barack sp. nov from Nihoa.

Our molecular characterizations of Ligia individuals collected in Nihoa show this population to be highly divergent and isolated from other Ligia lineages and species found in the HI. We observed no sharing of haplotypes between L. barack sp. nov individuals and other Ligia populations in the HI at any of the four mitochondrial genes studied (e.g., COI, Cyt-b, 16S and 12S rDNA). Instead, Nihoa specimens harbored unique and private haplotypes that form a well-supported monophyletic group that excludes all other Ligia species from the HI and that are highly divergent from other ones found to date in Hawaiian Ligia. COI K2P divergences between Nihoa Ligia and other Ligia species from Hawaii ranged from 3.0–17.8%, values that are similar to other amongst species comparisons (Table 2). Meanwhile, within species COI K2P divergences amongst L. barack sp. nov individuals ranged from 0.0–0.7%. Not surprisingly, the K/θ ratio between L. barack sp. nov and its sister taxon (L. hawaiensis; K/θ = 9.912), greatly exceeds the K/θ ratio of 4 at which there is a 95% probability that two separate species are being compared (Rosenberg, 2003).

The phylogenetic placement of L. barack sp. nov is of interest, as our analyses recovered with high support both the monophyly of this newly described species and its sister relationship to L. hawaiensis (Fig. 4). The latter is a coastal species of Ligia whose distributional range is thought to be limited to the island of Kaua‘ i, the closest island to Nihoa. These islands are separated by ∼240 km of open ocean and have never been connected. This suggests that oceanic dispersal led to the colonization of these islands by Ligia. Nihoa’s older age (7.5 My) suggests the ancestor to L. hawaiensis in Kaua‘ i may have originated from Nihoa; however, additional work is necessary to establish the origins of these species as back-dispersals appear to have occurred in Ligia from the HI.

Despite consisting of the Ligia from the two oldest islands in our analyses (Kaua‘ i: 5 My, Nihoa 7.5 My), the monophyletic group consisting of L. hawaiensis + L. barack sp. nov clade was not found in a basal position in any of our analyses (Fig. 3). Instead, the most basal clade in all analyses was one comprised of Ligia species found in Maui and Hawai‘ i, the two youngest islands in the archipelago (<1.5 My). These findings are consistent with previous studies of Ligia from the HI (Santamaria et al., 2013; Santamaria, 2019) and suggest that the evolution of Ligia in the region have been shaped by colonization, extinction, and back-dispersal events.

Our description of L. barack sp. nov from the island of Nihoa underscores the importance of molecular approaches in conservation efforts in the PNMN. Future studies of Ligia from other islands in the PNMN are likely to uncover additional highly divergent genetic lineages likely representing new species in need of description. These studies may also help further elucidate the evolutionary history of Ligia in the HI. Meanwhile, molecular characterizations of other poorly dispersing organisms may similarly uncover new species or genetic lineages in other taxa and thus increase our understanding of the biodiversity of these highly remote and isolated islands. Molecular tools may also aid in the monitoring of the spread of alien species, a critical threat to the fauna and flora of the PMNM (DeFelice et al., 1998; Selkoe, Halpern & Toonen, 2008). Ligia exotica has been shown to occur in Midway Atoll, an island within the PMNM (Santamaria et al., 2022). This species of Asian origin known to have been introduced to manmade coastal habitats around the world and is a potential competitor to endemic coastal Ligia (Hurtado et al., 2018). The use of genetic tools such as COI barcoding and eDNA may be useful to monitor the presence of this species in other regions of the PMNM without extensive field-work.

Conclusion

The use of both mitochondrial and nuclear gene fragments to characterize Ligia isopods from Nihoa uncovered a highly divergent lineage of Hawaiian Ligia not previously reported from other localities in the HI. Phylogenetic and species delimitation approaches provide evidence that this lineage represents a new species of Ligia, which we describe as Ligia barack sp. nov. To our knowledge, this species is the first intertidal crustacean that is described from and likely solely endemic to the island of Nihoa. This discovery underscores the unique biodiversity of the PMNM and the need for additional studies of poorly dispersing taxa within it. Our findings also further provide evidence of Ligia isopods as an example of in-situ speciation of a Hawaiian marine animal.

Supplemental Information

Supplemental Information 1 Concatenated alignment used for phylogenetic analyses and MSDAs

We would like to thank Pelikaokamanaoio “Pelika” Andrade for their help collecting specimens in Nihoa as well as students in the Santamaria laboratory at the University of Tampa for helping complete molecular work. We also would like to thank J. Hau‘ oli Lorenzo-Elarco and the members of the Papahānaumokuākea Native Hawaiian Cultural Working Groups Nomenclature Hui for bringing a wealth of traditional knowledge and insight to the naming process, resulting in a Hawaiian common name which recognizes the significance of Nihoa and perpetuates the genealogical relationship between Kānaka‘ Ōiwi (Native Hawaiians) and the environment.

Additional Information and Declarations

Competing Interests

Author Contributions

Field Study Permissions

DNA Deposition

Data Availability

New Species Registration

Carlos A. Santamaria is an Academic Editor for PeerJ.

Carlos A. Santamaria conceived and designed the experiments, performed the experiments, analyzed the data, prepared figures and/or tables, authored or reviewed drafts of the article, and approved the final draft.

Annabelle Bork performed the experiments, analyzed the data, prepared figures and/or tables, authored or reviewed drafts of the article, and approved the final draft.

Alexandra J. Larson performed the experiments, analyzed the data, prepared figures and/or tables, authored or reviewed drafts of the article, and approved the final draft.

Daniel J. Link conceived and designed the experiments, prepared figures and/or tables, authored or reviewed drafts of the article, conducted and organized sampling in Nihoa, and approved the final draft.

The following information was supplied relating to field study approvals (i.e., approving body and any reference numbers):

The collection of specimens from Nihoa was conducted under a permit granted to the Papahanaumokuakea Marine National Monument Co-Trustees, which include the U.S. Fish and Wildlife Service, by the State of Hawai‘ i Board of Land and Natural Resources (Permit Number PMNM-2022-001).

The following information was supplied regarding the deposition of DNA sequences:

The sequences obtained from the holotype are available at GenBank: PP851829 (COI); PP852382 (16S rDNA); PP852388 (12S rDNA); PP856001 (Cyt-b); PP852394; (28S rDNA); PP856007 (NaK); and PP861092 (H3A).

The following information was supplied regarding data availability:

The aligned dataset used for all analyses in this study is available in the Supplementary File.

The following information was supplied regarding the registration of a newly described species:

Publication LSID: urn:lsid:zoobank.org:pub:6CE79D26-19BA-435D-94A8-5A822ADD 42B0

Species name: urn:lsid:zoobank.org:act:558494AB-37D7-47BA-BA54-4E532D7585C6.

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
