# Peer review of "Description of a novel Ligia species from Nihoa, a remote island in the Papahānaumokuākea Marine National Monument"

_PeerJ, doi:10.7717/peerj.19373_

## Round 0.1 · original submission · Major Revisions

Dear Dr Santamaria and colleagues,

Many thanks for submitting your manuscript, which has now been reviewed by four reviewers. All reviewers offered a very positive feedback, praising not just the scientific content and rigour of the paper but also its clarity in presentation, structure and language. However, three of the four reviewers also recommend some revision. The main point that all three reviewers raise is the lack of description of the morphological description of the Ligia barack (including the morphological characteristics that distinguish this from other species) that can complement the genetic data.

Please note that all three reviewers have also attached documents with specific comments to the text.

Please make sure you address all reviewers' comments in your revision.

·

Basic reporting

No comment.

Experimental design

No comment.

Validity of the findings

No comment

Additional comments

The manuscript is written well. Sections are clear. Although drawings to show some morphological features of the taxa could have been useful, the descriptions based on genes are adequate. Well, done.

·

Basic reporting

No comment

Experimental design

No comment

Validity of the findings

No Comment

Additional comments

Dear Editor,
The manuscript “Description of a novel Ligia species from Nihoa, a remote island in the Papahānaumokuākea Marine National Monument (#109050)” presents a valuable and well-conducted study on Ligia isopods, describing a new species, Ligia barack sp. nov., from Nihoa Island in the Papahānaumokuākea Marine National Monument (PMNM). The work significantly advances our understanding of cryptic speciation in isolated ecosystems and highlights the importance of molecular taxonomic approaches in uncovering biodiversity.
The molecular analysis, which included sequences of four mitochondrial and three nuclear genes, provides robust evidence for species delimitation. The authors effectively integrate biogeographical and genetic data to make a compelling case for recognizing Ligia barack sp. nov. as a distinct lineage. Given the clarity of the findings and the ecological and conservation relevance of the study, I recommend the manuscript for acceptance with no revisions.

Reviewer 3 ·

Basic reporting

The paper is obvious and its structure is very nice, including the fluidity when it is read. The figures are of excellent quality and are self-explained. Some discussion could be improved with the papers of Dimitriou et al. (2022, 2023), Campos-Filho et al. (2023), Wang et al. (2022 and 2024), where cryptic taxa were recognized and/or described.

Dimitriou et al. 2022: https://doi.org/10.1016/j.ympev.2022.107585
Dimitriou et al. 2023: https://doi.org/10.1016/j.ympev.2023.107884
Campos-Filho et al. 2023: https://doi.org/10.11646/zootaxa.5270.1.3
Wang et al. 2022: https://doi.org/10.1186/s40850-022-00120-1
Wang et al. 2024: https://doi.org/10.3897/asp.82.e113041

Experimental design

The molecular delineation is correct and robust. However, the morphology was not attended at all, and as part of a species description, it must to include it.

Validity of the findings

The results bring out the authors' long-term work, as mentioned in the manuscript. As demonstrated in the distance table, the results show a high divergence between the taxa examined, even in standards not commonly observed to delimit species, but genera! In other words, the molecular validity of the work is high. However, the morphological work is weak, and the paper deals also with a species description. As recommended by ICZN, species descriptions must have illustrations of the types and characters that distinguish the species. Since the paper brings out a cryptic taxon, the most useful would be to produce illustrations or high-quality photos of the species, not just the dorsal habitus.

Annotated reviews are not available for download in order to protect the identity of reviewers who chose to remain anonymous.

Reviewer 4 ·

Basic reporting

i. The MS deals with the discovery of new species of Ligia, L. barack sp.nov. from PMNM. Owing to the importance of the study area, the work can add on to its diversity and can be a good example of evolutionary divergence.
ii. The MS is well structured mostly standing on the molecular experimental design.
iii. The tables and images provided are fitting to the journal criteria.
iv. The specimen deposition in zoological record and rules of ICZN are adhered looking to the data provided.
v. The MS is well structured and minor grammatic errors as marked in MS to be corrected.
vi. I commend the authors and appreciate the molecular inputs made to address the new finding. Also, the detailed methodology section is appreciated. Yet, there are some clarifications and inputs which I feel should be incorporated and justified in MS;
vii. I strongly feel that adding new sp. to science is important input and should be strongly supported by maximum evidences possible. Although being important, the present support largely includes only molecular data base. I think, authors should also include morphological data of the species as being novel. This can be some detailed/distinguishable characters (images/photo/line drawings). As this can aid other researchers in same line to compare. Morphological evidences always serve as primary confirmations.
viii. Moreover, kindly compare the new species with the closes relative and justify how it differs or resembles. Please check Line 259-260 which states there is lack of diagnostic differences from Hawaii????There are taxonomic papers to be studied where these differences are marked.
ix. Why only six species found from Nihoa are considered for molecular comparison, why not close resemblance from Hawaii?
x. Other minor corrections in MS attached.

Experimental design

I commend the authors and appreciate the molecular inputs made to address the new finding. Also, the detailed methodology section is appreciated.

As stated above, the morphological data needs to be added for new species.

Validity of the findings

The MS has provided the necessary data and sequences base whereever needed. The results and discussions are supported by molecular correlations and findings. The new species can be better validated adding morphological findings as earlier stated.
Discussion can be more strengthened using some more related references.

Additional comments

The MS can be considered for acceptance after the necessary revisions stated.

Annotated reviews are not available for download in order to protect the identity of reviewers who chose to remain anonymous.

---

## Round 0.2 · accepted · Accept

Dear Dr Santamaria and colleagues,

Many thanks for revising your manuscript. I have sent back your revision to one of the original reviewers. Following their comments and my own reading of your revised manuscript, I am happy to accept it in its current form. Congratulations!

Reviewer 3 ·

Basic reporting

The paper is clear and concise, and the text is fluid. The mentioned references are sufficient to support the introduction and discussion sections. The figures are of good quality and will help in future papers.

Experimental design

No comment.

Validity of the findings

The paper describes a new cryptic species of Ligia, contributing to a better understanding of species delimitations within Oniscidea. As in the first round, the paper has a robust methodology based on total evidence.